

# Recognition of species groups of *Naupactus* Dejean (Coleoptera: Curculionidae) from Argentina and neighboring countries

María G. del Río* and Analía A. Lanteri*

División Entomología, Museo de La Plata, FCNyM, CONICET, Universidad Nacional de La Plata, La Plata, Buenos Aires, Argentina
* These authors contributed equally to this work.

## ABSTRACT

*Naupactus* Dejean is the most diverse genus of the tribe Naupactini (Curculionidae: Entiminae), with more than 200 species occurring in South America, of which about 40 range in Argentina and neighboring countries. The Argentinean species treated herein were classified into nine groups having different biogeographic patterns: (1) the groups of *Naupactus xanthographus*, *N. delicatulus* and *N. auricinctus* mainly occur in northeastern Argentina (Misiones province) and reach the highest species diversity in the Atlantic and Parana forests of Brazil; (2) the groups of *N. hirtellus, N. cinereidorsum, N. rivulosus* and *N. tarsalis* show the highest species diversity in the Chacoan biogeographic province and also occur in the Yungas, Espinal, Monte, Parana forest (Argentina) and Cerrado (Brazil); (3) the groups of *N. leucoloma* and *N. purpureoviolaceus* have the highest species diversity in the Pampean biogeographic province, being also present in adjoining areas, mainly Chaco, Espinal, Monte and Parana forest. We provide descriptions, a dichotomous key, habitus photographs and line drawings of genitalia for the identification of the nine species groups, and a list of the Argentinean species from each group, together with their abbreviated synonymies, updated geographic distributions (including six new country records and several state/province records) and host plant associations. We discuss the characters that allow the separation of the species groups in a geographic distribution context, and provide information on species reassigned to genera other than *Naupactus;* among these, we transferred *N. cephalotes* (Hustache) to the tribe Tanymecini, genus *Eurymetopellus*, establishing the new combination *Eurymetopellus cephalotes*.

## INTRODUCTION

*Naupactus* Dejean is the most diverse genus of the tribe Naupactini (Curculionidae: Entiminae), with ca. 200 species occurring in South America (*Wibmer & O'Brien, 1986*; *Bordón, 1997*) and five in Central America, which were previously classified in

Corresponding authors
María G. del Río,
gdelrio@fcnym.unlp.edu.ar
Analía A. Lanteri,
alanteri@fcnym.unlp.edu.ar

*Alceis* Billberg by *O'Brien & Wibmer (1982)*. However, the precise number is still uncertain because new species remain to be described while others should probably be synonymized (*Lanteri & del Río, 2017a*).

*Hustache (1947)* published the only comprehensive taxonomic work on *Naupactus* from Argentina and neighboring countries, which included a dichotomous key, descriptions of several species and new geographic records for taxa previously described by *Boheman (1833)* and *Schoenherr (1840)* and by himself (*Hustache, 1923*, *1926*, *1938*). Nonetheless, these contributions lack illustrations of habitus or external features of taxonomic value, descriptions of genital characters, information on host plant associations and provide limited geographic data. The checklist of *Wibmer & O'Brien (1986)* included new synonymies proposed by Kuschel and updated country records for several species. They reported 45 Argentinean species, but this number was increased to 57 by *Morrone (1999)* as a result of transferring most of the South American taxa previously classified in *Pantomorus* Schoenherr to *Naupactus*.

The earliest classification of the South American *Naupactus* into species groups was made by *Lanteri & Marvaldi (1995)* and *Lanteri & del Río (2017a)*. They described and revised the *N. leucoloma* Boheman and *N. xanthographus* (Germar) species groups, which include some species of great economic importance due to the damage they cause to agriculture in Argentina and other countries worldwide (*Buchanan, 1939*; *Elgueta, 1993*; *Guzmán, Lanteri & Confalonieri, 2012*; *Lanteri et al., 2013*). Moreover, they analyzed the phylogenetic placement of *Naupactus* within the tribe Naupactini (*Lanteri & del Río, 2017b*; *del Río et al., 2018*) and proposed new synonymies for some species showing color variation, morphotypes and/or sexual dimorphism (*del Río & Lanteri, 2018*).

The main objectives of this paper are to recognize and describe the species groups of *Naupactus* distributed in Argentina and neighboring countries; to provide a dichotomous key for their identification; to illustrate the diagnostic features of these groups by habitus photographs and line drawings of genitalia; and to provide updated information on the synonymies, host plant associations and geographic distributions for all the Argentinean species included in these groups. The recognition of species groups is very useful to identify species belonging to highly diverse genera and helps to understand their evolution in a historical biogeographic context (*del Río, Morrone & Lanteri, 2015*; *del Río et al., 2018*).

## MATERIALS AND METHODS

This study is based on the examination of about 5,000 specimens deposited in the following entomological collections: American Museum of Natural History (New York, NY, USA), Arizona State University Charles W. O'Brien Collection (Tempe, AZ, USA), Coleção Entomologica do Instituto Oswaldo Cruz (Rio de Janeiro, Brazil), Departamento de Zoologia da Universidade Federal do Paraná (Curitiba, Brazil), Fundación e Instituto Miguel Lillo collection (San Miguel de Tucumán, Argentina), Museo de La Plata (La Plata, Argentina), Museu Nacional do Rio de Janeiro (Rio de Janeiro, Brazil), Museu de Zoologia da Universidade de São Paulo (São Paulo, Brazil), Muséum National d'Histoire Naturelle (Paris, France), The Natural History

Museum (London, UK), United States National Museum (Washington, DC, USA) and Universidad de la República (Montevideo, Uruguay).

Dissections of female and male genitalia were made according to standard entomological techniques (*Lanteri & O'Brien, 1990*). Genital characters were drawn using a camera lucida adapted to a Nikon SMZ800 stereoscopic microscope (Tokyo, Japan). An ocular micrometer attached to this microscope was used to take measurements.

The biogeographical scheme used to describe the distributions of the Naupactini genera are in agreement with that of *del Río, Morrone & Lanteri (2015)*, except for the Espinal (*sensu Cabrera & Willink, 1973*), which is here considered as separate from the Chacoan and Pampean biogeographic provinces. In Argentina, the Espinal is a xerophilous forest dominated by *Prosopis* L. that constitutes an arch between the latter biogeographic provinces.

The electronic version of this article in portable document format will represent a published work according to the International Commission on Zoological Nomenclature (ICZN), and hence the new names contained in the electronic version are effectively published under that Code from the electronic edition alone. This published work and the nomenclatural acts it contains have been registered in ZooBank, the online registration system for the ICZN. The ZooBank LSIDs (Life Science Identifiers) can be resolved and the associated information viewed through any standard web browser by appending the LSID to the prefix http://zoobank.org/. The LSID for this publication is: urn:lsid:zoobank.org:pub:B9BB18E6-2ED8-4928-9FA5-4B5CB2027BDF. The online version of this work is archived and available from the following digital repositories: PeerJ, PubMed Central and CLOCKSS.

## RESULTS

In the current contribution, we recognized 37 species of *Naupactus*, which were assigned to nine species groups ranging in Argentina, Bolivia, southern Brazil, Paraguay and Uruguay. These groups are characterized by a particular combination of characters of the external morphology and genitalia, and occur in one or more biogeographic provinces of the Neotropical Region.

### Key to species groups and subgroups of *Naupactus* from Argentina

1. Elytra with pair of tubercles at apex . . . . . . . . . . . . . . . . . . . . . . . . . . . . . . . . . . . . . . . . . . . . . . . . . . . . . . . . . . . . . . . . . . . . . . . . . . . . . . . . . . . . . **N. xanthographus** species group (Figs. 1A and 1B) . . .
1a. Elytra lacking pair of tubercles at apex . . . . . . . . . . . . . . . . . . . . . . . . . . . . . . . . . . . . . . . . . . 2
2. Humeri well-developed; base of elytra bisinuate. Pronotum usually subconical and smooth or slightly punctate. Metathoracic wings well-developed . . . . . . . . . . . . . . . . . . . . . . . 3
2a. Humeri usually reduced, if well-developed, elytral setae long and erect; base of elytra straight. Pronotum subcylindrical, usually granulose. Metathoracic wings reduced to absent . . . . . . . . . . . . . . . . . . . . . . . . . . . . . . . . . . . . . . . . . . . . . . . . . . . . . . . . . . . . . . . . . . . . . 7
3. Metatibial apex with broad, squamose corbel. Species usually more than 13 mm long . . . 4
3a. Metatibial apex lacking corbel or with very slender corbel. Species usually less than 13 mm long . . . . . . . . . . . . . . . . . . . . . . . . . . . . . . . . . . . . . . . . . . . . . . . . . . . . . . . . . . . . . . . . . 5

4. Antennae moderately robust; scape reaching to slightly exceeding hind margin of eye. Scaly vestiture dull-colored and uniformly distributed, or arranged in stripes along pronotum and elytra, leaving denuded areas. Disc of pronotum usually convex, with deep median groove. Intervals of elytra convex; punctures of striae usually broad . . . . . . . . . . . . . . . . . . . . . . . . . . . . . . . . . . . . . . . . . . . . . . . . . . . *N. rivulosus* species group (Figs. 1C–1E)

4a. Antennae slender; scape exceeding anterior margin of pronotum. Scaly vestiture completely covering integument. Disc of pronotum flat or impressed; median groove linear to indistinct. Intervals of elytra usually flat; punctures of striae small. . . . . . . . . . . . . . . . . . . . . . . . . . . . . . . . . . . . . . . . . . . . . . . . . . . . . . *N. delicatulus* species group (Figs. 1F and 1G)

5. Integument slightly sclerotized, covered with iridescent green, purple or copper colored scales; elytral setae usually long and erect. Antennae slender, usually reddish; scape reaching to slightly exceeding hind margin of eye. Pronotum subcylindrical, punctuate or rugose, usually very convex in males. Disc of elytra flat to slightly convex, lower than pronotum in lateral view. Spermathecal duct spiraled. . . . . . . . . . . . . . . . . . . . . . . . . . . . . . . . . . . . . . . . . . . . . . . . . . . . . *N. auricinctus* species group (Figs. 1H and 1I)

5a. Integument strongly sclerotized, uniformly covered with dull-colored scaly vestiture or with pair of yellow or green longitudinal stripes, leaving denuded areas; elytral setae short, recumbent. Antennae stout, black; scape not exceeding hind margin of eye. Pronotum usually subconical, smooth and flat. Disc of elytra convex, higher than pronotum in lateral view. Spermathecal duct not spiraled. . . . . . . . . . . . . . . . . . . . . . . . . . . . . . . 6

6. Scaly vestiture usually green, yellow or cream, arranged in stripes along pronotum and elytra, leaving remaining parts of integument denuded. Lateral carinae of rostrum sharp, denuded. Base of elytra usually strongly bisinuate; intervals slightly convex . . . . . . . . . . . . . . . . . . . . . . . . . . . . . . . . . . . . . . . . . . . . *N. tarsalis* species group . . . (Figs. 2A–2D)

6a. Scaly vestiture usually dull-colored, uniformly distributed or almost lacking. Lateral carinae of rostrum not sharp, squamose. Base of elytra slightly bisinuate; intervals usually flat . . . . . . . . . . . . . . . . . . . . . . . . . . . . . . . *N. cinereidorsum* species group (Figs. 2E and 2F)

7. Protibiae lacking row of denticles on inner edge. Metatibial apex with narrow, setose corbel. Pronotum almost smooth. Scaly vestiture iridescent green or gray; elytral setae of medium-length and suberect . . . . . . . . . . . . . . . . *N. hirtellus* species group (Figs. 3A and 3B)

7a. Protibiae with row of denticles on inner edge. Metatibial apex usually lacking corbel (external edge thickened/in some species). Pronotum granulose. Scaly vestiture dull-colored or almost lacking, accompanied by long, suberect to erect elytral setae, or iridescent green, purple or copper colored, having short, recumbent elytral setae . . . . . . . . . . . . . . . . . . . . . . . . . . . . . . . . . . . . . . . . . . . . . . . . . . . . . . . . . . . . . . 8

8. Scaly vestiture dull-colored, with characteristic lateral white stripes from apex of rostrum to apex of elytra, accompanied by long and erect elytral setae . . . . . . . . . . . . . . . . . . . . . . . . . . . . . . . . . . . . . . . . . . . . . . . . . . . . . *N. leucoloma* species group (Fig. 3C)

8a. Scaly vestiture completely lacking on dorsal surface, or iridescent green, purple or copper colored, accompanied by short, recumbent elytral setae or dull-colored, but without white stripes from apex of rostrum to apex of elytra . . . . . . . . . . . . . . . . . . . . . . . . . . . . . . . . . . . . . . . . . . . . . . . . . . *N. purpureoviolaceus* species group (Figs. 3D–3I)

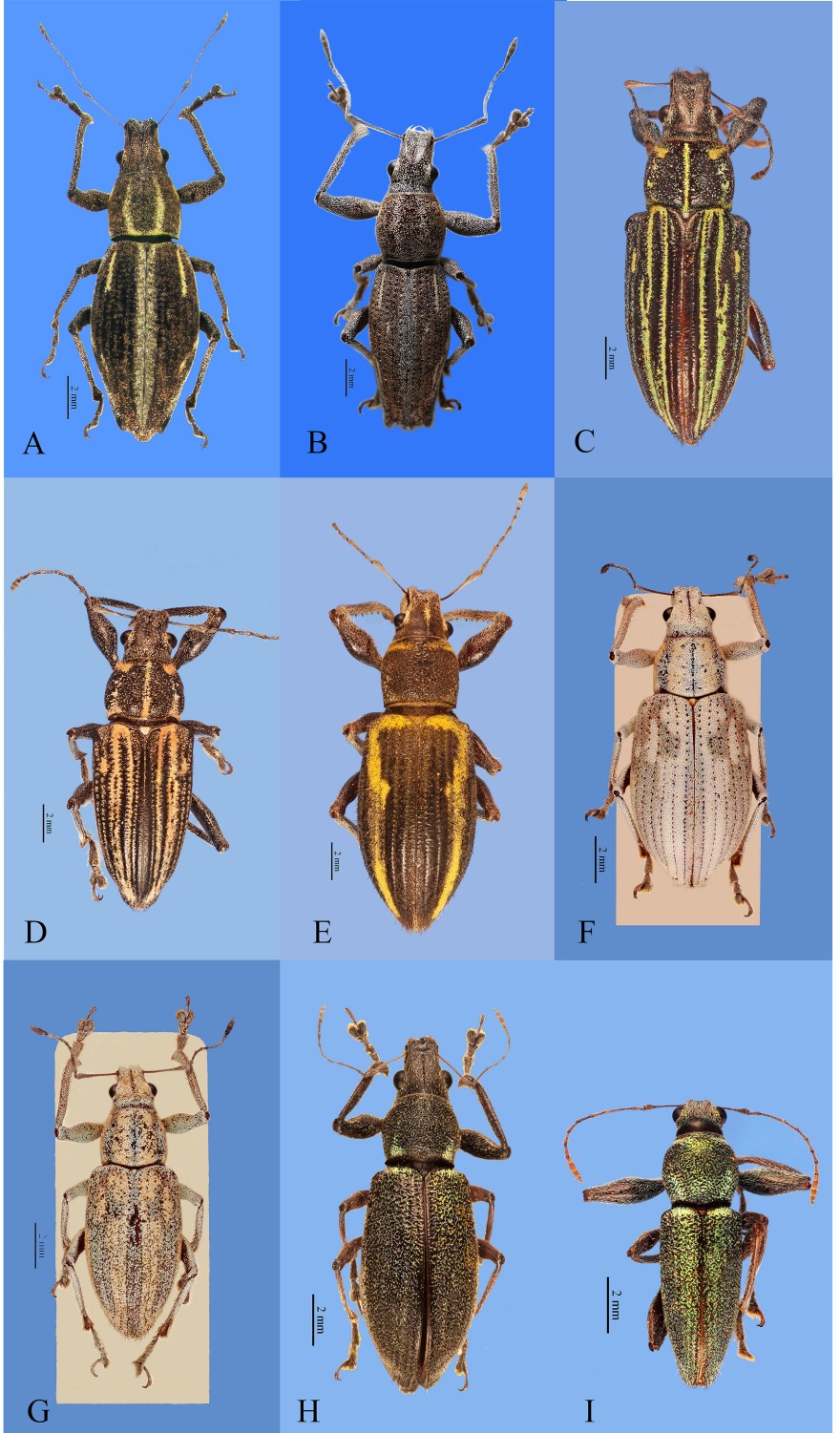

**Figure 1** **Habitus photographs of *Naupactus xanthographus*, *N. rivulosus* and *N. auricinctus* species groups, dorsal views.** (A) *Naupactus xanthographus*, female, (B) *N. xanthographus*, male, (C) *N. rivulosus*, female, (D) *N. rivulosus*, male, (E) *N. sulphurifer*, female, (F) *N. delicatulus*, female, (G) *N. delicatulus*, male, (H) *N. auricinctus*, female and (I) *N. auricinctus*, male. Credits: Bruno Pianzola.

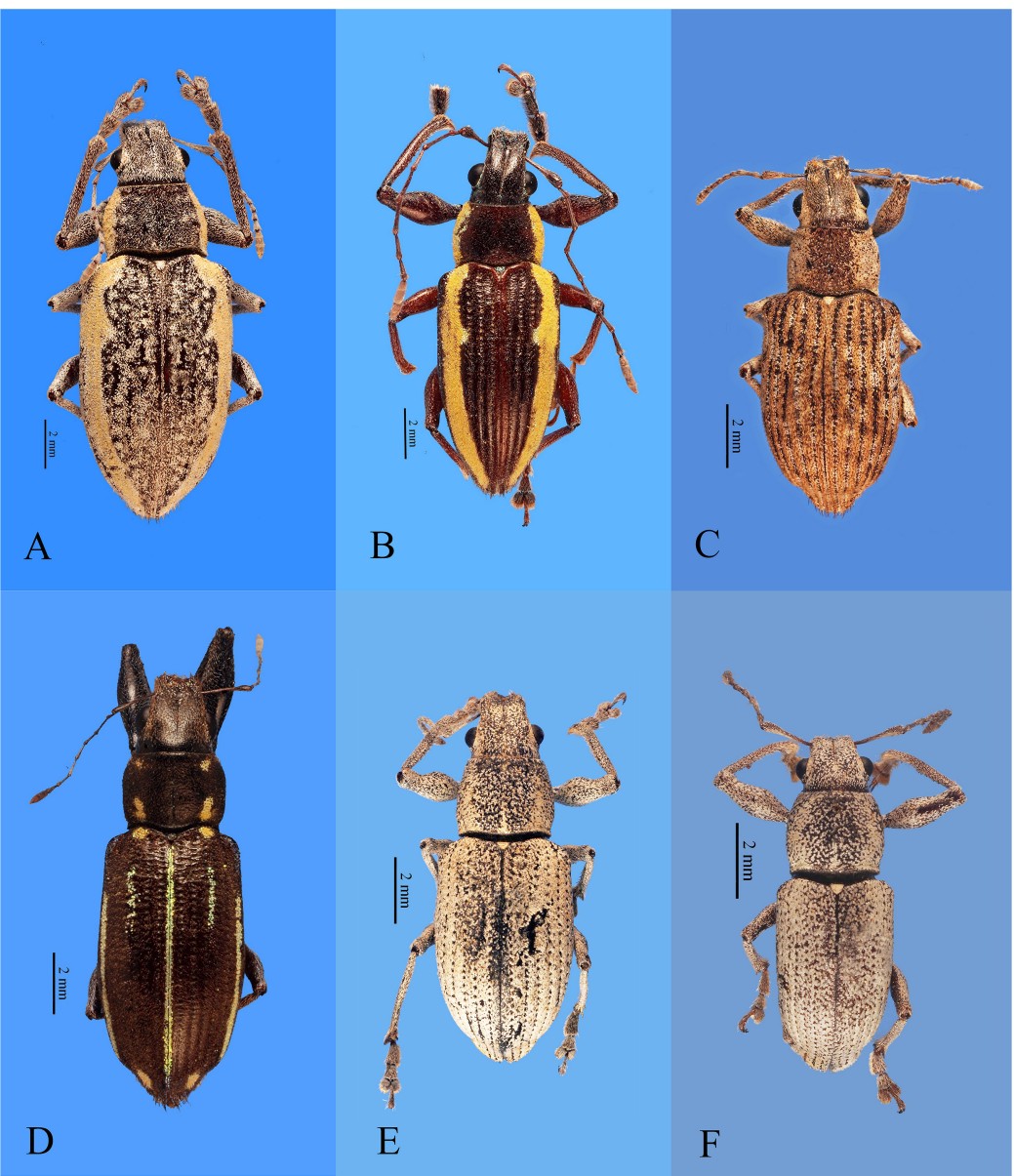

**Figure 2 Habitus photographs of *Naupactus tarsalis* and *N. cinereidorsum* species groups, dorsal views.** (A) *Naupactus tarsalis*, female, (B) *N. tarsalis*, male, (C) *N. cyphoides*, female, (D) *N. viridicinctus*, female, (E) *N. cinereidorsum*, female and (F) *N. cinereidorsum*, male. Credits: Bruno Pianzola.

## Criteria for inclusion of *Naupactus* species recorded from Argentina

The 37 species of *Naupactus* from Argentina under consideration show the combination of characters detailed in *del Río et al. (2018)*. This species number is the result of the following taxonomic and nomenclatural decisions:

1. *Lanteri & Marvaldi (1995)* synonymized *Graphognathus* Buchanan with *Naupactus*, and consequently the three species of "white-fringed beetles" assigned to *Graphognathus*

in *Wibmer & O'Brien (1986)* were named *N. leucoloma* Boheman, *N. minor* (Buchanan) and *N. peregrinus* (Buchanan).

2. Some species were recently transferred from *Naupactus* to other genera, for example, *N. carinirostris* (Hustache) and *N. signatus* (Blanchard) sensu *Morrone (1999)*, to *Parapantomorus* Emden, and to *Symmathetes* Schoenherr, respectively (*del Río & Lanteri, 2018*); *N. sulphureoviridis* (Hustache), originally classified in *Teratopactus* Heller (*del Río, Lanteri & Guedes, 2006*) and later in *Naupactus*, was recently placed in the original genus (*del Río & Lanteri, 2018*); and *N. inermis* Hustache was synonymized with *Macrostylus* (*Mimographus*) *ocellatus* Lanteri (*del Río & Lanteri, 2018*) and transferred to *Lanterius* Alonso-Zarazaga & Lyal, based on the results of a cladistic analysis (*del Río et al., 2018*).

3. *Lanteri & del Río (2017a)* described the new species *N. marvaldiae* for the *N. xanthographus* species group, and *del Río & Lanteri (2018)* established new synonymies for several species showing color variation, sexual dimorphism or polymorphism, that is, *N. bridgesii* G.R. Waterhouse (senior syn. of *N. angulithorax* Hustache); *N. auricinctus* (senior syn. of *N. ruficornis* Boheman); *N. peregrinus* (Buchanan) (senior syn. of *N. brevicrinitus* Hustache); *N. condecoratus* Boheman (senior syn. of *N. bosqi* Hustache); and *N. cyphoides* (Heller) (senior syn. of *N. calamuchitanensis* Hustache, *N. viridinitens* Hustache and *N. viridulus* Hustache).

4. *Naupactus transversus* Boheman and *N. ambiguus* Boheman were excluded from this paper because they only occur in Brazil. The specimens misidentified as *N. transversus* in most entomological collections probably belong to a new species (they were labeled as *N. missionum* Kuschel in litteris at the MZSP) distributed in Argentina, Brazil and Paraguay; and the specimens usually misidentified as *N. ambiguus* correspond to *Pantomorus postfasciatus* (Hustache), which share the geographic range of the latter species, plus Uruguay.

5. We excluded several species transferred from *Pantomorus* to *Naupactus* by *Morrone (1999)* because, in agreement with *Wibmer & O'Brien (1986)*, we consider that they should be better classified in *Pantomorus*, at least until a comprehensive revision of the South American species of this genus is made. These are: *Pantomorus auripes* Hustache, *Pantomorus fulvus* Hustache, *Pantomorus prasinus* Hustache, *Pantomorus similis* Hustache and *Pantomorus ruizi* (Brèthes), which are within the *Pantomorus auripes* species group (*Morrone & Lanteri, 1991*; *Lanteri et al., 1991*; *Lanteri, 1995*; *Scataglini, Lanteri & Confalonieri, 2005*); *Pantomorus viridisquamosus* (Boheman) and *Pantomorus obrieni* Lanteri, within the *Pantomorus viridisquamosus* species group (*Lanteri & Loiácono, 1990*); the probably related *Pantomorus humilis* Hustache and *Pantomorus hirsuticeps* Hustache; the highly modified *Pantomorus luteipes* Hustache, displaying ant mimicry; *Pantomorus bruneus* Hustache from northwestern Argentina, which is probably close to *Pantomorus minutus* Hustache and to *Pantomorus minutellus* Wibmer & O'Brien from Bolivia and Peru, and allied to *Asymmathetes* Wibmer & O'Brien; *Pantomorus postfasciatus* (Hustache) originally described in *Asynonychus* and frequently misidentified as

*N. ambiguus* Boheman (*Lanteri, Guedes & Parra, 2002*; *Guedes, Lanteri & Parra, 2005*); and the controversial *Pantomorus cinerosus* (Boheman) originally described in *Naupactus* and having six synonyms assigned to *Pantomorus* (*Hustache, 1947*; *Wibmer & O'Brien, 1986*; *Morrone, 1999*; *Lanteri, Marvaldi & Suárez, 2002*; *Guedes, Lanteri & Parra, 2005*).

6. We excluded the "fuller's rose weevil" *N. cervinus* Boheman because of its controversial taxonomic position (*del Río et al., 2018*). Indeed, it has been classified in *Asynonynchus* Crotch (*Hustache, 1947*; *Lanteri, 1986*; *Morrone, 1999*), in *Pantomorus* (*Buchanan, 1939*; *O'Brien & Wibmer, 1982*; *Wibmer & O'Brien, 1986*) and in *Naupactus* (*Alonso-Zarazaga & Lyal, 1999*). *Asynonychus* (type species *Asynonychus godmani* Crotch = *N. cervinus* Boheman) was synonymized with *Naupactus* in *Alonso-Zarazaga & Lyal (1999)*, but we consider that it should be treated either as a genus independent from *Naupactus* and *Pantomorus*, or as a junior synonym of *Alceis* Billberg.

7. *Asynonychus cephalotes Hustache, 1947*, classified in *Pantomorus* by *Wibmer & O'Brien (1986)* and in *Naupactus* by *Morrone (1999)*, it is not a Naupactini and meets definition of genus *Eurymetopellus* Emden, therefore it is herein transferred to Tanymecini, genus *Eurymetopellus*, thus establishing the new combination *Eurymetopellus cephalotes* (Hustache).

## Descriptions of groups of *Naupactus* from Argentina

### *Naupactus xanthographus* species group (Figs. 1A and 1B)

This group, which was revised by *Lanteri & del Río (2017a)*, includes six species mainly distributed in the Parana and Atlantic forests of Brazil. In Argentina, *N. dissimilis* Hustache, *N. navicularis* Boheman and *N. marvaldiae* Lanteri & del Río are restricted to Misiones province; *N. dissimulator* Boheman ranges from this province to Buenos Aires along the gallery forests of the Parana river, reaching the southernmost distribution limit of the species group in the Parana forest; and *N. xanthographus* (Germar) is the only species widely distributed in the Central and Northern areas of the country and has also been introduced in Chile (*Guzmán, Lanteri & Confalonieri, 2012*). *N. mimicus* Hustache is endemic to the Atlantic-Parana forests of Brazil. Within the *N. xanthographus* species group, *Lanteri & del Río (2017a)* recognized two subgroups, one including *N. xanthographus, N. dissimilis, N. navicularis* and *N. mimicus*, and the other *N. dissimulator* and *N. marvaldiae*.

**Species Included:**

*1-Naupactus dissimilis Hustache, 1947*

**Geographic distribution.** Northeastern Argentina (Misiones), Brazil (Paraná, Rio Grande do Sul and Santa Catarina), Paraguay (Alto Paraná, Caaguazú and Itapúa) and Uruguay (Artigas) (*Lanteri & del Río, 2017a*).

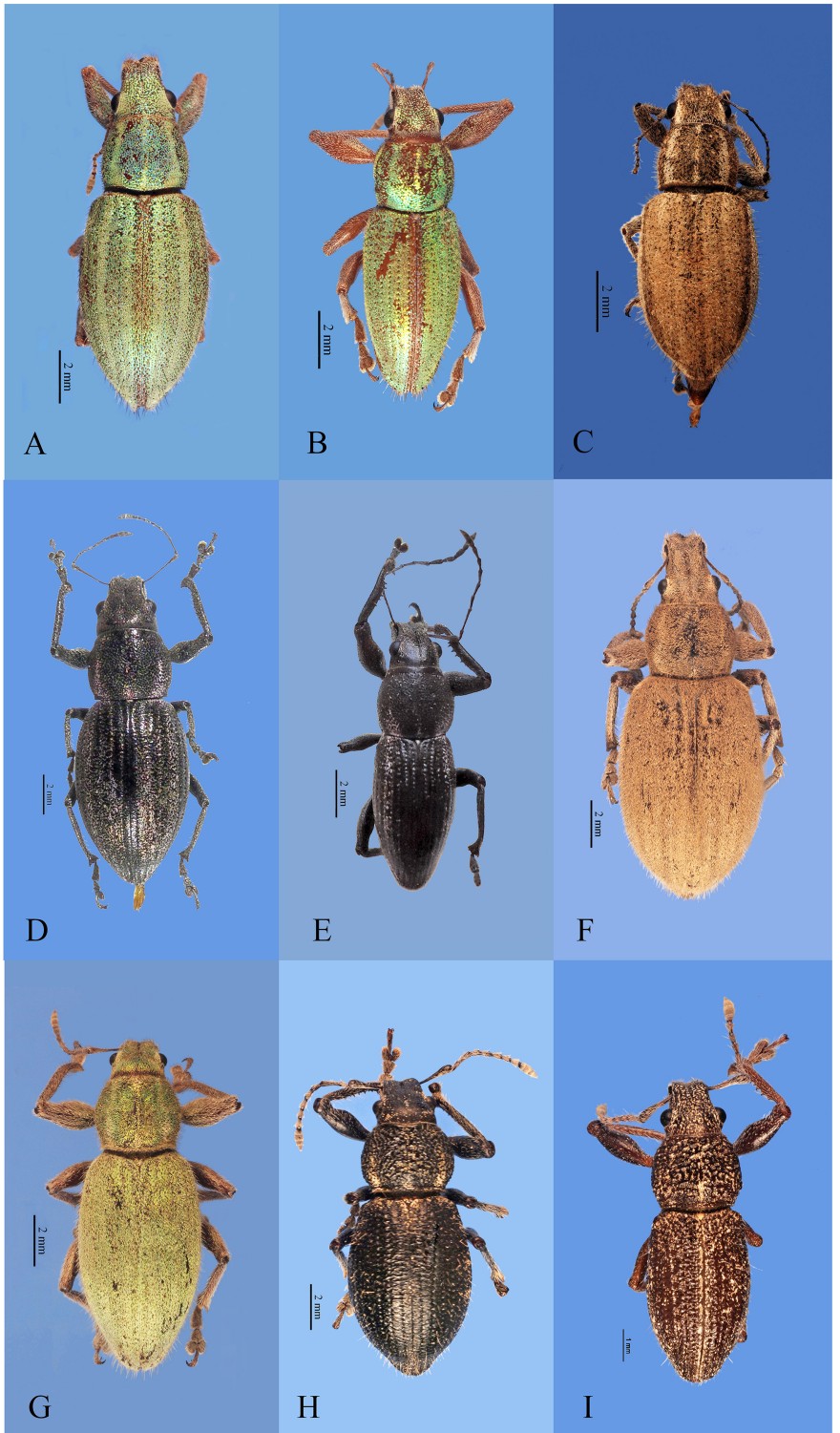

**Figure 3** **Habitus photographs of *Naupactus hirtellus*, *N. leucoloma* and *N. purpureoviolaceus* species groups, dorsal views.** (A) *Naupactus hirtellus*, female, (B) *Naupactus hirtellus*, male, (C) *N. tucumanensis*, female, (D) *N. purpureoviolaceus*, female, (E) *N. purpureoviolaceus*, male, (F) *N. chordinus*, female, (G) *N. chordinus*, male, (H) *N. rugosus*, female and (I) *N. rugosus*, male. Credits: Bruno Pianzola.

**Host plants.** *Coniza albida* Willd. ex Sprengel (Asteraceae), *Citrus maxima* (Burm.) Merr. (Rutaceae), *Ilex paraguariensis* Saint Hill. (Aquifoliaceae) and *Zea mays* L. (Poaceae) (*Lanteri & del Río, 2017a*).

*2-Naupactus dissimulator* **Boheman, 1840** (= *N. fallax* Boheman, 1840)

**Geographic distribution.** Argentina (Buenos Aires, Corrientes, Entre Ríos, Misiones and Santa Fe), Bolivia (Cochabamba), Southern Brazil (Paraná, Santa Catarina and Río Grande do Sul), Paraguay (Itapúa) and Uruguay (Colonia, Maldonado, Montevideo and Paysandú) (*Lanteri & del Río, 2017a*).

**Host plants.** *Ocotea punchella* (Nees & Mart.) Mez (Lauraceae), *Citrus sinensis* L. (Rutaceae), *I. paraguariensis* Saint Hill. (Aquifoliaceae) and *Prunus persicae* (L.) Batsch (Rosaceae) (*Lanteri & del Río, 2017a*).

*3-Naupactus marvaldiae* **Lanteri & del Río, 2017**

**Geographic distribution.** Northeastern Argentina (Misiones) and Southern Brazil (Santa Catarina) (*Lanteri & del Río, 2017a*).

*4-Naupactus navicularis* **Boheman, 1840**

**Geographic distribution.** Northeastern Argentina (Misiones), Brazil (Minas Gerais, Paraná, Rio de Janeiro, Rio Grande do Sul, Santa Catarina and São Paulo) and Paraguay (Itapúa) (*Lanteri & del Río, 2017a*).

**Host plants.** *Citrus* sp, especially *Citrus sinensis* L. (Rutaceae) (*Lanteri, Marvaldi & Suárez, 2002*).

*5-Naupactus xanthographus* **(Germar, 1824)** (Figs. 1A and 1B)

**Geographic distribution.** Argentina (Buenos Aires, Catamarca, Chaco, Córdoba, Corrientes, Entre Ríos, La Pampa, La Rioja, Mendoza, Misiones, Neuquén, San Juan, San Luis, Santa Fe, Santiago del Estero and Tucumán), southern Brazil (Rio Grande do Sul and Santa Catarina), Paraguay (Itapúa and Paraguarí) and Uruguay (Artigas, Colonia, Montevideo, Soriano and Treinta y Tres). It was introduced in Chile in 1942 (*Durán, 1944*) and is currently widespread from Atacama to Valparaiso and Juan Fernández Islands (*Lanteri & del Río, 2017a*).

**Host plants.** *Erythrina crista-galli* L. (Fabaceae), *Vitis vinifera* L. (Vitaceae) and other fruit plants (*Ripa, 1983*; *Elgueta, 1993*; *Lanteri & del Río, 2017a*).

*Naupactus rivulosus* species group (Figs. 1C–1E)

It includes six species mainly distributed in the Chacoan biogeographic province: *N. argentinesis* Hustache and *N. bruchi* (Heller) are endemic to Argentina; *N. rivulosus* (Olivier) and *N. variegatus* Hustache have been only recorded in the Parana forest of Argentina (Misiones) and are also widespread in Brazil and Paraguay; *N. bridgesii*

Waterhouse mainly occurs in the Yungas of Argentina and Bolivia, and *N. sulphurifer* Pascoe in the Monte and Espinal biogeographic provinces. The latter species *N. sulphurifer* (Fig. 1E) reaches the southernmost distribution of this group.

Other species probably belonging to the *N. rivulosus* species group are *N. sahlbergi* Boheman from Brazil (Federal District, Goiás, Minas Gerais and Paraná) and *N. peruvianus* Hustache from the Yungas of Bolivia and Peru.

**Description.** Species large (female 17–22 mm; male 14–16 mm) (Figs. 1C–1E). Integument strongly sclerotized. Scaly vestiture dull-colored and uniformly distributed or forming green, yellowish or pinkish stripes along pronotum and elytra and leaving denuded areas; scales round, small, slightly overlapped; elytral setae short, recumbent. Rostrum directed forward, 0.90–1.15X as long as wide at apex; lateral carinae subparallel, blunt. Forehead 1.15–1.20X as wide as rostrum at apex. Eyes convex. Antennae stout, medium length; scape clavate, usually not reaching hind margin of eye; funicle article 2, 2.5–3X as long as article 1; remaining funicle articles about 3X as long as wide; club 2.5–3X as long as wide. Pronotum subconical, 1.10–1.30X as wide as long (subcylindrical in *N. sulphurifer*); disc convex, usually slightly punctate; median sulcus usually deep; posterolateral angles impressed; base bisinuate, beveled. Scutellum squamose, white. Elytra 1.85–1.95X as long as wide; disc distinctly higher than pronotum in lateral view; humeri very broad; base bisinuate; intervals convex; punctures usually large; subapical calli distinct. Metathoracic wings usually well-developed. Procoxae lacking denticles; profemora wider than metafemora, particularly in males; protibiae with large hook-like mucro and row of small denticles on inner edge; meso and metatibiae lacking mucro in females, mesotibiae with small mucro in males. Metatibial apex with large squamose corbel; dorsal comb about as long as distal comb.

*Female terminalia*: sternite VIII suboval with or without apical prominence (Fig. 4A) or subrhomboidal (Fig. 4B); apodeme 2.5–3.5X as long as plate. Ovipositor (Fig. 5A) about 2/3 as long as of abdomen, with rows of long setae along posterior half, on external side of baculi. Spermatheca (Fig. 6A) subcylindrical, strongly sclerotized on proximal half (walls very thickened); collum very short, ramus indistinct to slightly developed. Spermathecal duct sclerotized, slender (narrower than opening of collum), about 3–6X as long as spermatheca.

*Male genitalia*: apex of median lobe subacute with small prominence at the tip (Fig. 7A), slightly recurved; apodemes about as long as, to slightly shorter than median lobe (see Fig. 7F). Endophallus without sclerites.

**Species included:**

**6-*Naupactus argentinensis* (Hustache, 1926)**

**Geographic distribution.** Endemic to Argentina (Catamarca, Misiones, Salta, Santiago del Estero and Tucumán).

**Host plants.** *Gossypium hirsutum* L. (Malvaceae) (*Lanteri, Marvaldi & Suárez, 2002*).
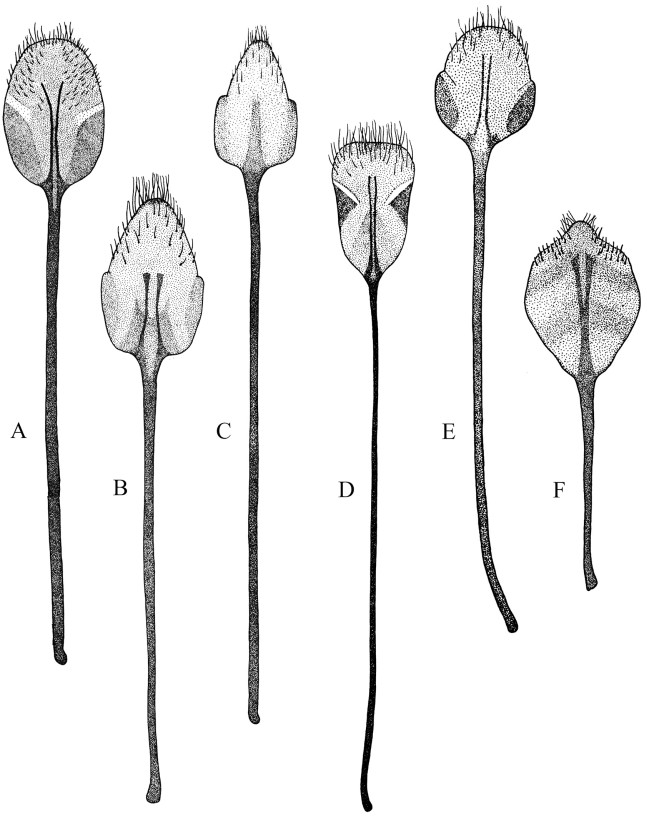

**Figure 4** **Female terminalia, sternite VIII.** (A) *Naupactus rivulosus*, suboval plate, (B) *N. argentinensis*, subrhomboidal plate, (C) *N. hirtellus*, subrhomboidal plate, (D) *N. tarsalis,* subpentagonal plate, (E) *N. denudatus*, subcircular plate and (F) *N. rugosus*, suboval plate with acute apex.

*7-Naupactus bridgesii* **Waterhouse, 1844** (= *N. angulithorax* *Hustache, 1947*)

**Geographic distribution.** Argentina (Catamarca, Formosa, La Rioja, Mendoza, Misiones, Río Negro, Salta, San Juan, San Luis and Tucumán) and Bolivia (Cochabamba, Potosí, Santa Cruz and Sucre).

**Host plants.** *Gossypium hirsutum* L. (Malvaceae) and *Medicago sativa* L. (Fabaceae) (*Lanteri, Marvaldi & Suárez, 2002*).

*8-Naupactus bruchi* **(Heller, 1921)** (= *Archopactus niveopectus* *Hustache, 1926*)

**Geographic distribution.** Endemic to Argentina (Salta, Santiago del Estero and Tucumán).

**Host plants.** *Prosopis kuntzei* Harms Kuntze (Fabaceae) (*Lanteri, Marvaldi & Suárez, 2002*).

*9-Naupactus rivulosus* **(Olivier, 1790)** (Figs. 1C–1D)

**Geographic distribution.** Northeastern Argentina (Misiones), Brazil (Amazonas, Bahía, Espirito Santo, Goiás, Mato Grosso do Sul, Minas Gerais, Paraná, Rio Grande do Sul, Rio

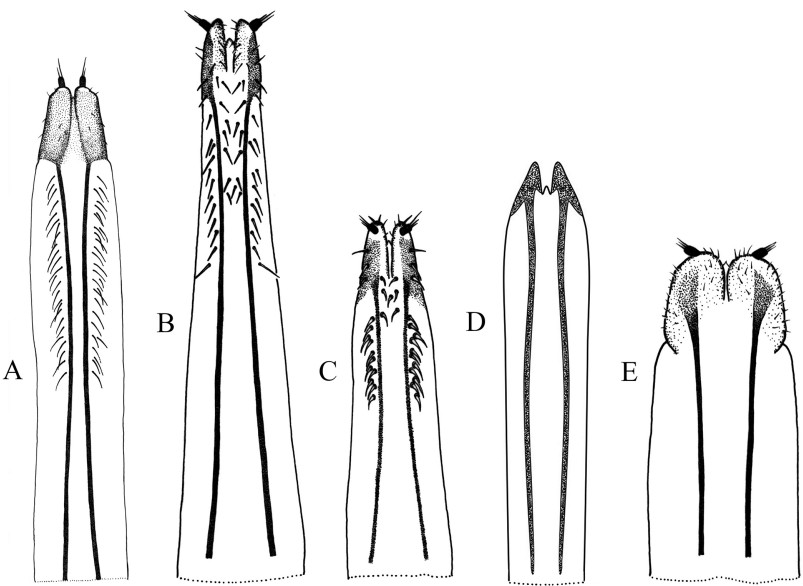

**Figure 5 Female genitalia, ovipositor, ventral views.** (A) *Naupactus argentinensis*, long, with rows of setae, (B) *N. chordinus*, long, with rows of setae, (C) *N. verecundus* moderately long, with rows of broad setae, (D) *N. dives*, moderately long, lacking setae and styli and (E) *N. rugosus*, short, lacking setae.

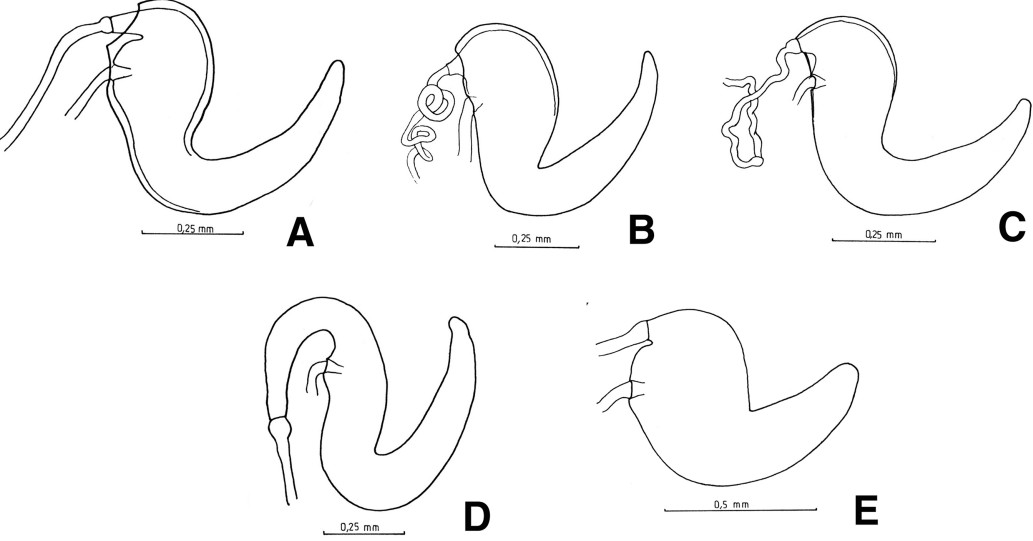

**Figure 6 Female genitalia, spermathecae.** (A) *Naupactus sulphurifer*, subcylindrical, (B) *N. auricinctus*, subcylindrical, duct spiraled, (C) *N. verecundus* subcylindrical, duct undulate, (D) *N. dissimulator*, subcylindrical with long collum and (E) *N. cyphoides*, subglobose.

de Janeiro, Santa Catarina and São Paulo) and Paraguay (Alto Paraná, Caazapá, Canindeyú, Guairá, Itapúa and Paraguarí).

**Host plants.** *Hibiscus* sp (Malvaceae) (Bosq, 1943), *Gossypium hirsutum* L. (Malvaceae), *V. vinifera* L. (Vitaceae), *Citrus* sp. (Rutaceae) and *Eucalyptus*

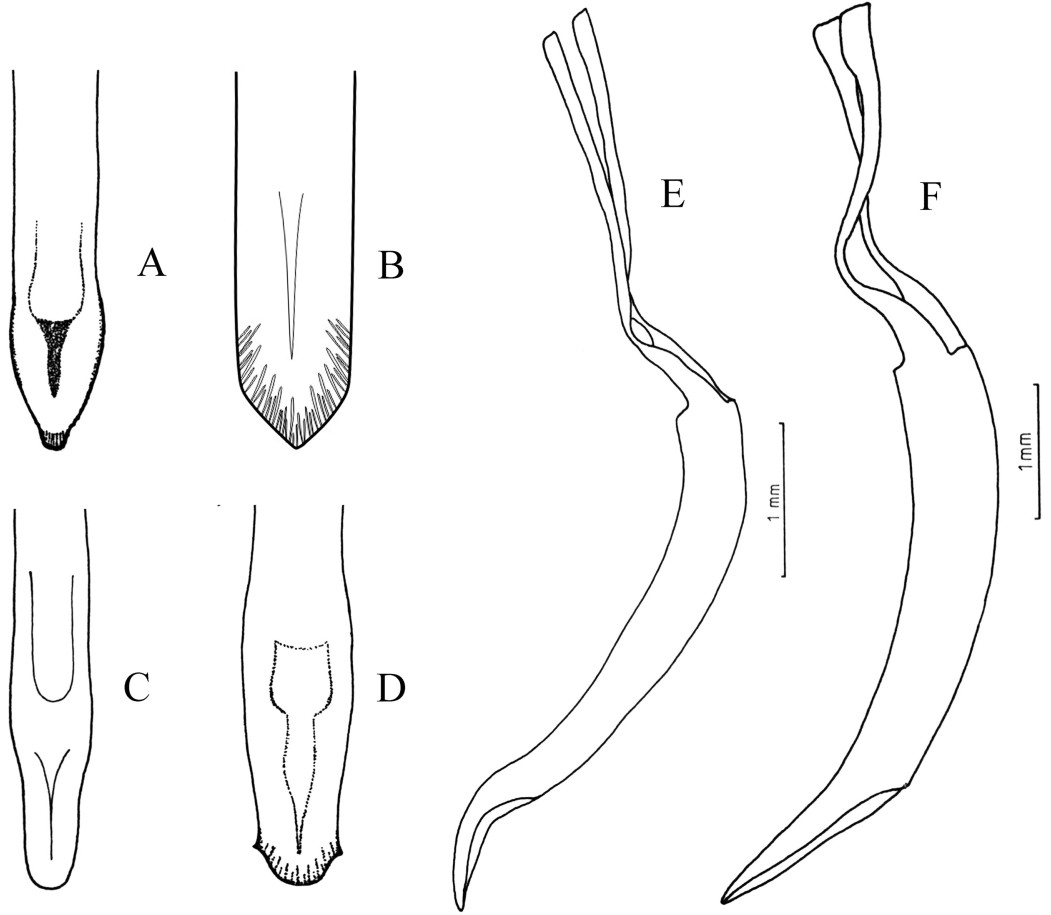

**Figure 7 Male genitalia, median lobe.** (A–D) Apex of median lobe, ventral views. (E and F) median lobe, lateral views. (A) *Naupactus sulphurifer,* subacute with prominence at the tip, (B) *N. delicatulus,* subtriangular, (C) *N. hirtellus,* rounded, (D) *N. xanthographus,* arrow-shaped, (E) *N. hirtellus,* apex strongly recurved and (F) *N. chordinus,* apex slightly recurved.

sp. (Myrtaceae) (*Silva et al., 1968*; *Lanteri, Marvaldi & Suárez, 2002*; *Lanteri, Guedes & Parra, 2002*).

**10-Naupactus sulphurifer Pascoe, 1881** (Fig. 1E)

**Geographic distribution.** Argentina (Buenos Aires, Catamarca, Córdoba, La Rioja, Mendoza, Misiones, Neuquén, Río Negro, Salta, San Luis, Santa Fe, Santiago del Estero and Tucumán), Paraguay, and Uruguay.

**Host plants.** *Larrea divaricata* Cav. and *Larrea cuneifolia* Cav. (Zygophyllaceae); *Gossypium hirsutum* L. and *Sphaeralcea* sp. (Malvaceae) (*Lanteri, Marvaldi & Suárez, 2002*).

**11-Naupactus variegatus Hustache, 1938**

**Geographic distribution.** Argentina (Misiones), Brazil (Paraná and Rio Grande do Sul) and Paraguay. Argentina and Paraguay are new country records.

*Naupactus delicatulus* **species group** (Figs. 1F and 1G)

It includes 15 species mainly occurring in the Parana and Atlantic forests of Brazil, only one recorded in Argentina, *N. delicatulus* Hustache (Misiones). The remaining species are: *N. ancora* Marshall, *N. balteus* Voss, *N. barbicauda* Boheman, *N. chalybeipes* Boheman, *N. deses* Schoenherr, *N. fuscus* Boheman, *N. leucographus* Boheman, *N. loripes* (Germar), *N. nubilosus* Boheman, *N. ochreonotatus* Voss, *N. pulchellus* Kuschel, N. *submaculatus Hustache, N. suturalis* Boheman and *N. wilsoni* Boheman. Some of these names may be synonyms.

**Description.** Species medium-sized (female 12–20 mm; male 11–16 mm) (Figs. 1F and 1G). Integument moderately sclerotized. Scaly vestiture very dense, brown or gray with cream of whitish markings; scales round, broadly overlapped; elytral setae short, recumbent. Rostrum directed forward, 1.15–1.30X as long as wide at apex; lateral carinae subparallel to slightly convergent anteriad. Forehead 1.25–1.35X as wide as rostrum at apex. Eyes convex. Antennae very slender, long; scape capitate, exceeding anterior margin of pronotum; funicular article 2, 2–2.45X as long as funicular article 1, remaining articles 3–3.70X as long as wide; club 3.10–4.10X as long as wide. Pronotum subconical, 1.15–1.35X as wide as long; disc flat to impressed, smooth to slightly undulate; base bisinuate. Scutellum squamose. Elytra 1.50–1.90X as long as wide, about as high as to slightly higher than pronotum in lateral view; humeri well-developed; base bisinuate; intervals flat to slightly convex; punctures small, with a scale or seta on bottom; subapical calli distinct only in females. Metathoracic wings well-developed. Legs long, slender; procoxae without denticles; profemora slightly wider than metafemora; protibiae with sharp, hook-like mucro and row of small denticles on inner margin; meso and metatibiae lacking mucro and denticles in females; mesotibiae with small mucro in males. Metatibial apex with broad, squamose corbel; dorsal comb longer than distal comb.

*Female terminalia*: sternite VIII subrhomboidal; apodeme about 2X as long as plate (see Fig. 4B). Ovipositor (see Fig. 5B) about 2/3 as long as abdomen, with rows of long setae along posterior half, on external side of baculi (see Fig. 5B). Spermatheca (Fig. 6D) subcylindrical, slightly sclerotized toward proximal end; collum conical, long; ramus distinct. Spermathecal duct sclerotized, wide, curled, about 10X as long as spermatheca (*Cyrtomon* like, see *Lanteri & del Río, 2016*).

*Male genitalia*: apex of median lobe triangular (Fig. 7B), slightly recurved; apodemes about as long as median lobe. Endophallus with sclerites consisted on a pyriform piece flanked by two wing-like pieces (*Cyrtomon*-like, see *Lanteri & del Río, 2016*).

**Species included:**

*12-Naupactus delicatulus* Hustache, 1947 (Figs. 1F and 1G)

**Geographic distribution.** Northeastern Argentina (Misiones) and southern Brazil (Rio Grande do Sul and Santa Catarina).

*Naupactus auricinctus* species group (Figs. 1H and 1I)

It includes about 20 species mainly distributed in the Atlantic and Parana forests of Brazil, three of them occurring in northeastern Argentina, Bolivia and Paraguay. The species recorded in Argentina are *N. auricinctus* Boheman, *N. condecoratus* Boheman and *N. versatilis* Hustache, being the latter the most widespread in the country. The other species that belong to this group are *N. aerosus* Boheman, *N. agglomeratus* Hustache, *N aurolimbatus* Boheman, *N. basalis* Hustache, *N. bipes* (Germar), *N. bondari* Marshall, *N. decorus* (Fabricius), *N. habenatus* Marshall, *N. hypocrita* (Germar), *N. jacobi* Hustache, *N. macilentus* Boheman, *N. parallelus* Hustache, *N. pedestris* Voss, *N. pithecius* (Germar), *N. proximus* Voss, *N. sanguinipes* Hustache, *N. univittatus* Boheman, *N. viridicyaneus* Hustache, *N. viridisquamosus* Boheman. Some of these species names may be synonyms.

**Description.** Species medium-sized to large (females 9–17 mm; males 8.5–12 mm) (Figs. 1H and 1I). Integument slightly sclerotized; antennae, tibiae and meso and metafemora often reddish. Scaly vestiture usually iridescent green, pinkish, copper-colored or grayish, in some species with lighter stripes along sides of pronotum and elytra; scales round, slightly overlapped; elytral setae fine, dark, erect, medium to very long. Rostrum directed forward, 1–1.30X as long as wide at apex; lateral carinae subparallel, usually sharp. Forehead 1.25–1.45X as long as rostrum at apex. Eyes usually convex. Antennae very slender, long; scape clavate to slightly capitate, usually exceeding hind margin of eye (except in *N. versatilis*); funicular article 2, 2.3–3.4X as long as article 1; remaining articles 3.35–4.7X as long as wide; club 3–3.5X as long as wide. Pronotum subcylindrical, often with remarkable sexual dimorphism, 1.20–1.35X as wide as long in females, 1.05–1.20X in males; disc slightly convex in females and strongly convex in males of some species, granulose to strigose; base usually straight, not beveled. Scutellum setose. Elytra 1.6–1.8X as long as wide, disc slightly higher than pronotum (females) or completely flat (males); humeri broad to slightly reduced; base straight to slightly bisinuate, not beveled; intervals flat, often with transversal rugosities; punctures small to indistinct; subapicall calli indistinct. Metathoracic wings well-developed. Procoxae often with denticles in males; profemora much wider than metafemora in males or some Brazilian species (*N. condecoratus*, *N. bipes*, *N. pithecius*); protibiae often inwardly curved, with large mucro and row of usually small denticles on inner margin; meso and metatibiae lacking mucro in females, and with small mucro in males. Metatibial apex without corbels; dorsal comb about as long as distal comb.

*Female terminalia:* sternite VIII (see Fig. 4B) subrhomboidal; apodeme 2–3.5X as long as plate. Ovipositor (see Fig. 5B) about 2/3 as long as abdomen, with rows of long setae along posterior half, on each side of baculi. Spermatheca (Fig. 6B) subcylindrical, strongly sclerotized toward proximal half; collum and ramus indistinct. Spermathecal duct (Fig. 6B) sclerotized, slender, usually spiraled.

*Male genitalia:* apex of median lobe subacute (see Fig. 7A), with or without distinct prominence at the tip; apodemes shorter or about as long as median lobe. Endophallus without sclerites.

**13-*Naupactus auricinctus* Boheman, 1833** (= *N. ruficornis* Boheman, 1840; *N. chloris* Hustache, 1947) (Figs. 1H and 1I)

**Geographic distribution.** Argentina (Misiones), Bolivia (Santa Cruz), Brazil (Mato Grosso, Mato Grosso do Sul, Minas Gerais, Paraná, Rio de Janeiro, Rio Grande do Sul, Santa Catarina and São Paulo) and Paraguay (Central, Cordillera, Guairá, Itapúa and Paraguarí).

**Host plants.** *Ilex paraguariensis* Saint Hill. (Aquifoliaceae) (Bosq, 1943) and *Gossypium hirsutum* L. (Malvaceae) in Argentina; *Pinus elliottii* Engelm. and *Pinus taeda* L. (Pinaceae); and *Casearia sylvestris* Sw. (Salicaceae) in Brazil.

**14-*Naupactus condecoratus* Boheman, 1840** (= *N. bosqi* Hustache, 1947)

**Geographic distribution.** Argentina (Corrientes and Misiones), Brazil (Paraná, Rio de Janeiro, Santa Catarina and São Paulo) and Paraguay (Itapuá).

**Host plants.** *Ilex paraguariensis* Saint Hill. (Aquifoliaceae) (Lanteri, Marvaldi & Suárez, 2002).

**15-*Naupactus versatilis* Hustache, 1947** (= *N. imbellis* Hustache, 1947)

**Geographic distribution.** Argentina (Chaco, Córdoba, Corrientes, Entre Ríos, Formosa, Misiones, Salta, Santa Fe, Santiago del Estero, and Tucumán), Brazil (Goiás, Minas Gerais, Paraná, Rio de Janeiro, Santa Catarina and São Paulo) and Paraguay (Guairá and Itapúa).

**Host plants.** *Citrus* sp. (Rutaceae) (Lanteri, Guedes & Parra, 2002).

***Naupactus tarsalis* species group** (Figs. 2A–2D)

It includes six Argentinean species mainly distributed in the Chacoan biogeographic province, two of them endemic, *N. hirsutus* Hustache and *N. laticollis* Hustache. *N. cyphoides* (Heller) (Fig. 2C) is the most widespread in Argentina and also ranges in Bolivia, Paraguay and Uruguay; *N. viridicinctus* Hustache (Fig. 2D) occurs in northeastern Argentina (Misiones) and the Parana forest of southern Brazil; *N. tarsalis* Boheman and *N. schapleri* Hustache are distributed in the Chacoan biogeographic province, the Yungas of Bolivia and the Cerrado in Brazil. Other species belonging to this group are *N. fatuus* Boheman, *N. lar* (Germar) and *N. roscidus* Erichson, mainly distributed in Brazil (Cerrado biogeographic province).

**Description.** Species medium sized to large (female 10–17 mm; male 8–13 mm) (Figs. 2A–2D). Integument strongly sclerotized. Scaly vestiture green, yellow or cream, forming stripes along pronotum and elytra, and leaving remaining parts of integument denuded, or dull-colored and uniformly distributed; scales, round, slightly overlapped

(piliform in *N. hirsutus*); elytral setae short, slender, recumbent (except *N. hirsutus*). Rostrum directed forward, 0.80–0.95X as long as wide at apex; lateral carinae convergent anteriad, sharp (subparallel and blunt in *N. viridicinctus*). Forehead 1.25–1.50X as wide as rostrum at apex. Eyes strongly to slightly convex. Antennae slender, medium length to long (stout and short in *N. cyphoides*); scape slightly capitate, almost reaching hind margin of eye; funicular article 2, 1.50–2.70X as long as article 1; remaining articles about 3.20–5X as long as wide (1.5X in *N. cyphoides*); club 2.5–4X as long as wide. Pronotum slightly subconical, 1.45–1.65X as wide as long; disc flat to slightly convex, smooth; median sulcus usually indistinct; postero-lateral angles slightly impressed; base bisinuate, beveled. Scutellum squamose or setose. Elytra 1.55–1.95X as long as wide, distinctly higher than pronotum in lateral view; humeri very broad; base bisinuate; intervals usually flat; punctures small; subapical calli slight. Metathoracic wings well-developed. Procoxae lacking denticles; profemora usually slightly wider than metafemora; protibiae with small acute mucro (large in *N. viridicinctus*) and row of small denticles on inner margin; meso and metatibiae lacking mucro and denticles. Metatibial apex lacking corbel (slender and setose in *N. cyphoides*); dorsal comb about as long as distal comb.

*Female terminalia*: sternite VIII suboval (see Fig. 4A), subpentagonal (Fig. 4D) or subromboidal (see Fig. 4B); apodeme about 3.25–4X as long as plate. Ovipositor (see Fig. 5B) about 2/3 as long as abdomen, with rows of long setae along posterior half, on external side of baculi. Spermatheca usually subcylindrical (see Fig. 6A) (subglobose in *N. cyphoides*; Fig. 6E) and strongly sclerotized toward proximal end (walls very thickened); collum indistinct; ramus indistinct to slightly developed. Spermathecal duct slightly sclerotized, usually more slender than opening of collum, short (1–3X as long as spermatheca).

*Male genitalia*: apex of median lobe subacute, curved, with small prominence at the tip (see Fig. 7A); apodemes about as long as, to slightly longer than median lobe. Endophallus without sclerites.

**Species included:**

*16-Naupactus cyphoides* **(Heller, 1921)** (Fig. 2C) (= *N. griseomaculatus* (*Hustache, 1923*); *N. prasinus Hustache, 1947*; *N. viridimarginalis Hustache, 1947*; *N. calamuchitanensis Hustache, 1947*; *N. viridinitens Hustache, 1947*; *N. viridulus Hustache, 1947*).

**Geographic distribution.** Argentina (Buenos Aires, Catamarca, Chaco, Chubut, Córdoba, Corrientes, Formosa, Misiones, La Pampa, La Rioja, Rio Negro, Salta, San Luis, Santa Fe, Santiago del Estero and Tucumán), Bolivia (Tarija and Santa Cruz), Paraguay (Alto Paraguay, Capital district, Central, Itapuá and Presidente Hayes) and Uruguay (Rio Negro).

**Host plants.** *Prosopis kutzei* Harms Kuntze, *Prosopis nigra* (Grisebach) Hieron, *Medicago sativa* L. (Fabaceae); *Schinopsis balansae* Engl. (Anarcadiaceae) (*Lanteri, Marvaldi & Suárez, 2002*).

**17-*Naupactus hirsutus* *Hustache, 1947***

**Geographic distribution.** Endemic to northwestern Argentina (Jujuy, Salta, Santiago del Estero and Tucumán).

**Host plants.** *Prosopis kutzei* Harms Kuntze, *Prosopis nigra* (Grisebach) Hieron, *Medicago sativa* L. (Fabaceae); *Schinopsis lorentzii* (Grisebach) Engler (Anarcadiaceae) (*Lanteri, Marvaldi & Suárez, 2002*).

**18-*Naupactus laticollis* *Hustache, 1947***

**Geographic distribution.** Endemic to northwestern Argentina (Salta and Santiago del Estero).

**Host plants.** *Prosopis nigra* (Grisebach) Hieron (Fabaceae) (*Lanteri, Marvaldi & Suárez, 2002*).

**19-*Naupactus schapleri* *Hustache, 1947***

**Geographic distribution.** Argentina (Formosa and Tucumán), Brazil (Mato Grosso, Mato Grosso do Sul and São Paulo) and Paraguay (Capital district and Itapúa). Brazil is a new country record.

**20-*Naupactus tarsalis* Boheman, 1840** (Figs. 2A and 2B) (=*N. glaucivittatus* Blanchard, 1847; *N. albidiventris* *Hustache, 1947*)

**Geographic distribution.** Argentina (Chaco, Corrientes, Formosa, Misiones, Salta, Santiago del Estero and Tucumán), Bolivia (Beni, Santa Cruz and Tarija), Brazil (Bahia, Federal District, Goiás, Maranhão, Mato Grosso, Mato Grosso do Sul, Minas Gerais, Pará, Pernambuco, Rio Grande do Norte, Rio de Janeiro, São Paulo and Tocantins) and Paraguay (Amambay, Concepción and Paraguarí).

**Host plants.** *Citrus* sp (Rutaceae) (*Lanteri, Guedes & Parra, 2002*)

**21-*Naupactus viridicinctus* (Fig. 2D) *Boheman, 1833***

**Geographic distribution.** Argentina (Misiones) and Brazil (Minas Gerais, Santa Catarina and São Paulo).

**_Naupactus cinereidorsum_ species group** (Figs. 2E and 2F)

It includes five Argentinean species mainly distributed in the Chacoan biogeographic province, none of them endemic to the country. *N. cinereidorsum* Hustache is the most widespread in Argentina and follows the arch of the Espinal; *N. castaneus* Hustache and *N. denudatus* Hustache range in Chaco and Yungas; *N. argentatus* Hustache is mainly distributed in Chaco, and *N. virens* Boheman is the only restricted to the northeast (Parana forest) and widespread in Brazil (Parana forest and Cerrado). Other species probably belonging to this group are *N. latifrons* Boheman, *N. morio* Boheman, *N. pictus* Boheman, from Brazil and *N. praedatus* Erichson, from Peru.

**Description.** Species medium-sized (female 11–14 mm; male 9–11 mm) (Figs. 2E and 2F). Integument strongly sclerotized. Scaly vestiture gray, pale green or pale brown, lacking stripes of different colors or almost denuded of scales on dorsum; scales round, slightly overlapped or lacking; elytral setae short, recumbent. Rostrum slightly directed forward, 0.85–1.10X as long as wide at apex; lateral carinae slightly convergent toward apex, blunt. Forehead 1.20–1.45X as wide as rostrum at apex. Eyes convex. Antennae moderately stout, medium length; scape clavate, not reaching to slightly exceeding hind margin of eye; funicular article 2, 2–2.35X as long as funicular article 1, remaining articles 2–3.8X as long as wide; club 2.85–4.10X as long as wide. Pronotum subcylindrical to slightly subconical, 1.20–1.50X as wide as long; disc flat, smooth to slightly granulose; base slightly bisinuate, beveled. Scutellum squamose. Elytra 1.50–1.90X as long as wide, slightly higher than pronotum in lateral view; humeri broad; base slightly bisinuate; intervals usually flat; punctures usually large; subapical calli indistinct. Metathoracic wings usually well-developed. Procoxae without denticles; profemora slightly wider than metafemora; protibiae with mucro and row of denticles on inner margin; meso and meta tibiae lacking mucro and denticles in females; metatibiae with small mucro in males. Metatibial apex without corbel; dorsal comb about as long as distal comb.

*Female terminalia*: sternite VIII subcircular (Fig. 4E) or subrhomboidal (see Fig. 4B); apodeme about 2–4X as long as plate (the shortest apodemes are recorded for *N. virens* and *N. argentatus*). Ovipositor (see Fig. 5A) 1/2 to 3/4X as long as abdomen, with rows of long fine setae along posterior half, on external side of baculi (more sparse and slender in short ovipositors). Spermatheca (see Fig. 6A) usually subcylindrical and strongly sclerotized toward proximal end (subglobose en *N. denudatus*, see Fig. 6E); collum very short; ramus usually indistinct. Spermathecal duct slightly sclerotized, slender (about same width of opening of collum), not spiralized, usually short (2–3X as long as spermatheca).

*Male genitalia*: apex of median lobe acute to subacute, with small prominence at the tip (see Fig. 7A); apodemes about as long as median lobe. Endophallus without sclerites.

**Species included:**

*22-Naupactus argentatus* Hustache, 1947

**Geographic distribution.** Argentina (Corrientes, Misiones, Santa Fe and Santiago del Estero) and Paraguay (Presidente Hayes). Paraguay is a new country record.

**Host plants.** *Saccharum officinarum* L. (Poaceae).

*23-Naupactus castaneus* Hustache, 1947

**Geographic distribution.** Argentina (Chaco and Misiones), Bolivia (Santa Cruz, Tarija) and Brazil (Mato Grosso do Sul). Brazil is a new country record.

*24-Naupactus cinereidorsum Hustache, 1947* (Figs. 2E and 2F)

**Geographic distribution.** Argentina (Buenos Aires, Chaco, Córdoba, Entre Ríos, Misiones, Santa Fe, Santiago del Estero and Tucumán) and Uruguay (Montevideo).

**Host plants:** *Medicago sativa* L. and *Glycine max* (L.) Merr. (Fabaceae), *Heliantus annuus* L. (Asteraceae) (*Lanteri, Marvaldi & Suárez, 2002*).

*25-Naupactus denudatus Hustache, 1947*

**Geographic distribution.** Argentina (Catamarca, Jujuy, Salta, Santiago del Estero and Tucumán) and Bolivia (Tarija).

*26-Naupactus virens Boheman, 1840*

**Geographic distribution.** Northeastern Argentina (Misiones), Brazil (Federal District, Goiás, Mato Grosso, Mato Grosso do Sul, Minas Gerais, Paraná, Rio de Janeiro, Rondonia and São Paulo) and Paraguay (Presidente Hayes). Paraguay is a new country record.

*Naupactus hirtellus* **(Voss) species group** (Figs. 3A and 3B)

It includes one species distributed in the Chacoan biogeography province of Argentina and Paraguay, *N. hirtellus*, and two species from the Yungas of Bolivia and Peru (*N. lizeri* Hustache and *N. viridimicans* Hustache).

**Description.** Species medium sized (females 12–14.5 mm; males 9–11 mm) (Figs. 3A and 3B). Integument slightly sclerotized. Scaly vestiture dense, iridescent green or opaque gray, except the usually denuded elytral suture; scales round to oval; elytral setae medium length, erect. Rostrum slightly directed forward (gular angle almost 90°), 1–1.15X as long as wide at apex; lateral carinae subparallel to slightly convergent anteriad, blunt. Forehead 1.350–1.50X as wide as rostrum at apex. Eyes convex. Antennae slender, medium length; scape clavate, reaching to slightly exceeding hind margin of eye; funicular article 2, 2.25–2.50X as long as funicular article 1; remaining articles 2.85–3X as long as wide; club oval, 3–3.6X as long as wide. Pronotum subcylindrical to slightly subconical, with slightly curved flanks, 1.15–1.35X as wide as long; disc flat, smooth to slightly granulose; base straight to slightly bisinuate. Scutellum squamose or setose. Elytra 1.50–1.65X as long as wide, slightly higher than pronotum in lateral view; humeri well-developed to slightly reduced, rounded; base straight to slightly bisinuate, not beveled; intervals flat; punctures usually indistinct; apical calli indistinct. Metathoracic wings usually well-developed. Procoxae without denticles; profemora slightly wider than metafemora; protibiae with small mucro and lacking row of denticles on inner margin; meso and metatibiae lacking mucro and denticles. Metatibial apex with narrow setose corbel; dorsal comb about as long as distal comb.

*Female terminalia*: sternite VIII subrhomboidal (Fig. 4C); apodeme 3–4X as long as plate. Ovipositor (see Fig. 5B) as long as abdomen, with rows of long setae along posterior half, on external side of baculi. Spermatheca subcylindrical (see Fig. 6A), moderately

sclerotized; collum and ramus indistinct. Spermathecal duct sclerotized, very slender, about 5X as long as spermatheca.

*Male genitalia*: apex of median lobe (Fig. 7C) very elongate, rounded, strongly recurved (Fig. 7E); apodemes slightly shorter than median lobe. Endophallus without sclerites.

**Species included:**

***27-Naupactus hirtellus* (Voss, 1932)** (=*N. cupreata* (Voss, 1932); *N. caroli* Hustache, 1947)

**Geographic distribution**. Argentina (Chaco, Jujuy, Formosa, Misiones and Salta) and Paraguay (Central, Concepción, Cordillera, Guairá and Paraguarí).

**Host plants.** *Gossypium hirsutum* L. (Malvaceae) in Paraguay.

***Naupactus leucoloma* species group** (Fig. 3C)

It was revised by *Lanteri & Marvaldi (1995)* and includes five species mainly distributed in the Pampean biogeographic province and nearby areas (Chaco, Yungas, Espinal and borders of the Paraná forest). *N. leucoloma* Boheman shows the broadest distribution (*Guzmán, Lanteri & Confalonieri, 2012*), being introduced in Chile (including Eastern Island and Juan Fernández Islands) and other countries worldwide. *N. minor* (Buchanan) and *N. peregrinus* (Buchanan) are typical Pampean species and were also introduced outside their native range. *N. albolateralis* Hustache is endemic to Argentina (Chacoan biogeographic province) and *N. tucumanensis* Hustache ranges also in Bolivia (Yungas) and Paraguay (Chaco).

**Species included:**

***28-N. albolateralis* Hustache, 1947**

**Geographic distribution.** Endemic to Argentina (Santiago del Estero).

***29-Naupactus leucoloma* Boheman, 1840** (= *N. dubius* (Buchanan, 1942), *N. fecundus* (Buchanan, 1947), *N. imitator* (Buchanan, 1947), *N. pilosus* (Buchanan, 1942), *N. striatus* (Buchanan, 1942).

**Geographic distribution.** Species broadly distributed in Argentina (Buenos Aires, Catamarca, Chaco, Chubut, Córdoba, Corrientes, Entre Ríos, Formosa, Jujuy, La Pampa, La Rioja, Mendoza, Río Negro, Salta, San Juan, San Luis, Santa Fe, Santiago del Estero and Tucumán), southern Brazil (Rio Grande do Sul) and Uruguay (Artigas, Canelones, Cerro Largo, Colonia, Durazno, Lavalleja, Montevideo, Paysandú, Soriano and Tacuarembó). Introduced in Chile, Peru, Mexico, Australia, New Zealand, South Africa and USA.

**Host plants.** There are about 385 known hosts for this species, including ornamentals, fruit trees, horticultural and industrial crops and forage. According to *Kuschel (1972)* it shows preferences for legumes, particularly *Phaseolus vulgaris* (Fabaceae). In Argentina the

main hosts of economic importance are alfalfa (*Medicago sativa* L.) and soybean (*Glycine max* L. (Merr.) (Fabaceae), strawberry (*Fragaria* sp) and *Prunus avium* L. (Rosaceae), onion (*Allium cepa* L.) (Amarydillaceae), potatoes (*Solanum tuberosum* L.) and pepper (*Capsicum annuum* L.) (Solanaceae) (*Lanteri, Marvaldi & Suárez, 2002*; *Lanteri et al., 2013*). *Solidago chilensis* Meyen and *Wedelia glauca* (Ort.) Hoff. (Asteraceae) are common wild host plants.

**30-*Naupactus minor* (Buchanan, 1947)** (= *N. breviscapus* *Hustache, 1947*)

**Geographic distribution.** Argentina (Buenos Aires, Entre Ríos and Santa Fe), southern Brazil (Rio Grande do Sul) and Uruguay (Canelones, Montevideo, Paysandú, Rivera). Introduced in USA.

**Host plants.** It is harmful for alfalfa (*Medicago sativa* L.), *Lotus* spp. and soybean (*Glycine max* L. (Merr.)) in Argentina, Brazil and Uruguay (*Lanteri, Marvaldi & Suárez, 2002*; *Lanteri et al., 2013*).

**31-*Naupactus peregrinus* (*Buchanan, 1939*)** (= *N. brevicrinitus* *Hustache, 1947*)

**Geographic distribution.** Argentina (Chaco, Entre Ríos, Misiones, Santiago del Estero and Santa Fe), southern Brazil (Rio Grande do Sul), Paraguay (Central) and Uruguay (Paysandú). Introduced in USA and Azores Islands.

**Host plants.** It is harmful for alfalfa (*Medicago sativa* L.) and soybean (*Glycine max* L. (Merr.)) in Argentina, Brazil and Uruguay (*Lanteri, Marvaldi & Suárez, 2002*; *Lanteri et al., 2013*). In Paraguay is associated with cotton, *Gossypium hirsutum* (Malvaceae).

**32-*N. tucumanensis* *Hustache, 1947*** (Fig. 3C)

**Geographic distribution.** Argentina (Buenos Aires, Catamarca, Chaco, Córdoba, Formosa, Salta, Santiago del Estero and Tucumán), Bolivia (Santa Cruz) and Paraguay (Itapúa).

**Host plants.** *Medicago sativa* L. (Fabaceae) (*Lanteri, Marvaldi & Suárez, 2002*), *Z. mays* L. (Poaceae) and *Schinopsis balansae* Engl. (Anarcadiaceae).

***Naupactus purpureoviolaceus* species group** (Figs. 3D–3I)

It includes five species mainly distributed in the Pampean biogeographic province. *N. rugosus* Hustache and *N. verecundus* Hustache are endemic to Argentina (the latter was introduced in Chile) and mainly occur in Espinal and Monte provinces; *N. chordinus* Boheman and *N. dives* (Klug) are typical Pampean species and *N. purpureoviolaceus* Hustache is broadly distributed in northeastern areas of Argentina, southern Brazil, Paraguay and Uruguay (Pampean, Chacoan and edges of Parana forest).

Other species that probably belong to this group are *N. alternevittatus* Hustache, *N. pilipes* Hustache and *N. termolerasi* Hustache, from Uruguay and southern Brazil.

**Description.** Species medium-sized to large (females 11–19 mm; males 9–17 mm) (Figs. 3D–3I). Integument moderately to strongly sclerotized. Scaly vestiture usually

sparse, iridescent green, purple or copper-colored, dull-colored or completely lacking on dorsal surface (*N. rugosus*); elytral setae short and recumbent or long and erect. Rostrum slightly directed forward (gular angle almost 90°), 1.10–1.20X as long as wide at apex; lateral carinae subparallel, very sharp (convergent anteriad and blunt in *N. rugosus*). Forehead 1.30–1.40X as wide as rostrum at apex. Eyes usually convex. Antennae stout, short to medium length; scape clavate, not reaching hind margin of eye; funicular article 2, 1.30–1.90X as long as funicular article 1; remaining articles 2–2.4X as long as wide; club 2.50–3.25X as long as wide. Pronotum subcylindrical, 1.25–1.45X as wide as long, with slightly curved and slightly granulose, or with strongly curved flanks and granulose; base straight, slightly beveled. Scutellum setose (squamose in *N. rugosus*). Elytra 1.45–170X as long as wide, higher than pronotum in lateral view; humeri reduced, rounded; base straight; intervals flat or slightly convex; punctures small; apical calli indistinct. Metathoracic wing reduced. Procoxae without denticles; profemora slightly wider than metafemora; protibiae with small to medium sized mucro and row of denticles on inner margin; meso and metatibiae usually lacking mucro and denticles (mesotibiae with mucro and denticles in males of *N. rugosus*). Metatibial apex without corbel; dorsal comb about as long as, to slightly shorter than distal comb.

*Female terminalia*: sternite VIII usually subrhomboidal (see Fig. 4B) (suboval with prominence at the tip in *N. rugosus,* Fig. 4F); apodeme usually 2.5–4X as long as plate (about 1.5X in *N. rugosus*). Ovipositor usually half length of abdomen, with rows of long and often coarse setae along posterior half, on external side of baculi (Figs. 5B and 5C) (setae indistinct in *N. dives* and *N. rugosus* Figs. 5D and 5E); coxites strongly sclerotized and styli lacking only in *N. dives* (Fig. 5D). Spermatheca (Fig. 6C) subcylindrical, strongly sclerotized on proximal half; collum and ramus indistinct. Spermathecal duct (Fig. 6C) usually slender, undulate and of variable length (1.5–4X as long as spermatheca).

*Male genitalia*: apex of median lobe usually subacute with small prominence at the tip (see Fig. 7A) (with broad prominence in *N. dives*); apodemes slightly shorter to about half length of median lobe (Fig. 7F). Endophallus without sclerites or with small U-shaped sclerites.

**Species included:**

**33-*Naupactus chordinus* Boheman, 1833** (Figs. 3F and 3G) (= *N. suffitus Boheman, 1833*; *N. fernandezi Hustache, 1947*; *N. sericellus Hustache, 1947*; *N. sericeus Hustache, 1947*).

**Geographic distribution.** Argentina (Buenos Aires, Misiones and Santiago del Estero) and Uruguay (Canelones, Cerro Largo, Montevideo and San José).

**Host plants.** *Glycine max* L. (Merr.) (Fabaceae), in Buenos Aires province.

**34-*Naupactus dives* Klug, 1829** (= *N. klugii Boheman, 1833*)

**Geographic distribution.** Argentina (Buenos Aires), Brazil (Rio Grande do Sul and Santa Catarina) and Uruguay (Lavalleja, Maldonado and San José).

**Host plants.** *Eryngium* sp (Apiaceae), a perennial plant introduced from Europe and currently extended in the Pampean biogeographic province.

*35-Naupactus purpureoviolaceus Hustache, 1947* (Figs. 3D and 3E) (= *N. persimilis* (*Hustache, 1947*))

**Geographic distribution.** Argentina (Buenos Aires, Chaco, Córdoba, Corrientes, Entre Ríos, Misiones, Santa Fe and Tucumán), Brazil (Río Grande do Sul), Paraguay (Alto Paraná, Caaguazí, Central, Concepción, Cordillera, Itapúa, Paraguarí and San Pedro) and Uruguay (Artigas, Cerro Largo, Maldonado, Paysandú, Rivera and Treinta y Tres). Brazil is a new country record.

**Host plants.** *Gossypium hirsutum* L. (Malvaceae) in Paraguay; *Phaseolus* sp. and *Glycine max* L. (Merr.) (Fabaceae), in southern Brazil.

*36-Naupactus rugosus Hustache, 1947* (Figs. 3H and 3I)

**Geographic distribution.** Endemic to central-western Argentina (Catamarca, La Pampa, Mendoza, San Luis and Santiago del Estero).

**Host plants.** *Eupatorium* sp., *Senecio subulatus* D. Don ex Hook. & Arn. and *Grindelia chiloensis* (Cornel) Cabrera (Asteraceae); *Larrea divaricata* Cav and *Larrea nitida* Cav. (Zygophyllaceae) (*Lanteri, Marvaldi & Suárez, 2002*).

*37-Naupactus verecundus Hustache, 1947* (= *N. calamuchitanus Hustache, 1947*; *N. vianai Hustache, 1947*)

**Geographic distribution.** Endemic to Argentina (Buenos Aires, Catamarca, Chubut, Córdoba, Corrientes, La Pampa, La Rioja, Mendoza, Neuquén, Río Negro, San Luis, Santiago del Estero and Tucumán) and introduced in Chile. *Silva et al. (1968)* cited this species for southern Brazil, associated with grapes, however, we could not confirm the presence of *N. verecundus* in this country.

**Host plants.** *Baccharis salicifolia* (Ruiz et Pavón) Pers., *B. spartioides* (Hook. et Arn.) Remy. (Asteraceae) and *Portieria* sp (Zygophyllaceae). It is harmful for *V. vinifera* L. (Vitaceae), other fruit plants and ornamental garden shrubs in Argentina (Mendoza province) and central Chile (*Elgueta, 1993*; *Lanteri, Marvaldi & Suárez, 2002*).

## DISCUSSION AND CONCLUSIONS

We recognized nine species groups of *Naupactus* showing different biogeographic patterns which are consistent with the results of a track analysis reported by *del Río, Morrone & Lanteri (2015)*. The *N. xanthographus*, *N. delicatulus* and *N. auricinctus* species groups mainly occur in the Atlantic and Parana forests of Brazil, and most of the Argentinean species are only recorded in the northeast of the country (Misiones province). The *N. hirtellus, N. cinereidorsum, N. rivulosus* and *N. tarsalis* species groups have the highest species diversity in the Chacoan biogeographic province, with some of them also being present in neighboring biogeographic provinces (Yungas, Espinal, Pampa,

Parana forest and Cerrado in Brazil). The *N. leucoloma* and *N. purpureoviolaceus* species groups are mainly distributed in the Pampean biogeographic province, and also occur in nearby areas, for example, Chaco, Espinal and Monte. Some species of these groups reach the southernmost distribution limit of the genus *Naupactus* in the Americas, and exhibit morphological characters typical of treeless and/or xerophilous environments, for example, reduction of hind wings, a strongly sclerotized integument almost denuded of scales, presence of long erect setae on the elytra, and presence of rows of denticles along the inner edge of all tibiae. In all *Naupactus* groups, the most differentiated species occur at the edges of their geographic distributions, where the environmental conditions are usually more extreme e.g. *N. rugosus* in the Monte biogeographic province; *N. suphurifer* mainly in Monte and Espinal, and *N. dives* in Espinal and arid areas of the Pampean biogeographic province.

Some species groups are more homogeneous than others in terms of external morphology and/or characters of genitalia. The species included in the group of *N. rivulosus* (type species of *Naupactus*) share quite uniform ovipositors and spermathecae, for example, ovipositor about 2/3 as long as the abdomen, with rows of setae along the posterior half and well-developed styli (*del Río et al., 2018*) (Fig. 5A), and spermatheca with strongly sclerotized walls, a very short collum (duct lobe) and a slightly developed ramus (gland lobe) (Fig. 6A). The sternite VIII varies in shape, being oval with or without apical prominence, or subrhomboidal as in most Naupactini (*Lanteri & del Río, 2017b*) (see Figs. 4A, 4B and 4F), and usually bears a long apodeme. *N. sulphurifer* (Fig. 1E) has a female terminalia typical of the *N. rivulosus* species group but differs in some characters of external morphology, for example, a subcylindrical and slightly granulose pronotum instead of a subconical and smooth pronotum as in most species of the group, slightly longer antennae than those of the remaining species, and all tibiae having rows of denticles on the inner edge, at least in males.

*Naupactus cyphoides* is the most different within the *N. tarsalis* species group (Fig. 2C), showing shorter antennae than the remaining species (the funicle article 2 is less than two times as long as the funicle article 1 and the funicle articles 3–7 are less than two times as long as wide at apex); a pronotum more than 1.50 times as wide as long; a metatibial apex with a very slender corbel; and a broader spermatheca without thickened walls (Fig. 7E). The spermatheca of the single Argentinean species of the *N. delicatulus* species group does not show the typical generic characters, for example, it has a subcylindrical and very long collum, a well-developed ramus (see Fig. 6D) and a broad and curled spermathecal duct resembling that of the species of *Cyrtomon* Schoenherr (*Lanteri & del Río, 2016*). A spermatheca of similar shape is observed in *N. dissimulator* and *N. marvaldiae*, both belonging to the same subgroup within the *N. xanthographus* species group.

The *N. auricinctus* species group is quite homogeneous in terms of external morphology and genitalia and differs from the other groups by the presence of a spiraled spermathecal duct (Fig. 6B). Based on a phylogenetic analysis by *del Río et al. (2018)*, this species group may be related to the *N. purpureoviolaceus* species group, which is characterized by having a usually undulate spermathecal duct (Fig. 6C). Two species of the latter group exhibit ovipositor characters that differ from those of most *Naupactus*. In *N. dives*,

the ovipositor is distinguished by lacking long setae on its posterior half, by having more sclerotized coxites as compared to the remaining species and by lacking styli (Fig. 5D); while *N. rugosus* has a short and wide ovipositor without the typical rows of long setae (Fig. 5E). The ovipositor of *N. dives* resembles that of *Floresianus* Hustache, *Eurymetopus* Schoenherr and *Priocyphus* Hustache, all of which seem to be adapted for laying eggs into the soil (*Lanteri & del Río, 2008*); and the ovipositor of *N. rugosus* resembles that of *Trichonaupactus densior* Hustache, both of which inhabit similar environments (*Lanteri & del Río, 2008*, *2016*).

The sternite VIII of *N. rugosus* looks like that of some species within the *N. rivulosus* species group (e.g., *N. bruchi*), but it has a shorter apodeme (Fig. 4F). Moreover, in *N. rugosus* the integument is more strongly sclerotized, the scaly vestiture is almost lacking, the pronotum is more rugose and all the tibiae have rows of denticles on the inner edge, particularly in males. Likewise, other Naupactini from environments that are also dominated by xerophilous trees and thorny shrubs (i.e., *Enoplopactus* Heller, *Priocyphopsis* Hustache, *Mendozella* Hustache) have denticles on all tibiae and often display tubercles on pronotum and elytra (*Lanteri, 1990*; *Lanteri & del Río, 2016*).

The morphology of the male genitalia provides few useful characters for the differentiation of species groups of *Naupactus*. In most species, the apex of the median lobe is subacute and ends in a narrow or broad prominence (Figs. 7E–7F). In the *N. leucoloma* and *N. purpureoviolaceus* species groups, this prominence may be indistinct and the apodemes are usually much shorter than the median lobe (Fig. 7F), whereas in most of the remaining groups the latter are usually about as long as the median lobe (Fig. 7E). The arrow-shaped (Fig. 7D) or subtriangular (Fig. 7B) apex, recorded for some members of the *N. xanthographus* species group (*N. xanthographus*, *N. dissimilis* and *N. mimicus*) and *N. delicatulus*, respectively, are not typical of *Naupactus*. In addition, only *N. delicatulus* and two species of the *N. xanthographus* species group (*N. dissimulator* and *N. marvaldiae*) have distinct sclerites in the endophallus, which consist of a pyriform piece flanked by two wing-like pieces (*Cyrtomon*-like, see *Lanteri & del Río, 2017b*); this feature correlates with a spermatheca of different shape (i.e., with a long subcylindrical collum).

A phylogenetic analysis by *del Río et al. (2018)*, suggests that the *N. leucoloma* species group is probably monophyletic and closely related to the *N. purpureoviolaceus* and *N. auricinctus* species groups; the *N. tarsalis* species group (represented by *N. cyphoides*) may be close to the *N. cinereidorsum* species group; and the two subgroups of the *N. xanthographus* species group do not form a clade. Even if future studies reveal that some species groups of *Naupactus* herein described are not natural, we believe that their recognition is useful in facilitating species identification, and in testing phylogenetic hypotheses in a historical biogeographic context (see *del Río, Morrone & Lanteri, 2015*; *del Río et al., 2018*).

## ACKNOWLEDGEMENTS

Thanks are due to all the curators for loaning material and helping us with the collections, to Bruno Pianzola for taking the photographs, to Paulina Hernandez for technical assistance and to Silvia Pietrovsky for English revision of the manuscript.

### Funding

This work was supported by the Consejo Nacional de Investigaciones Científicas y Técnicas (CONICET), Agencia Nacional de Promoción Científica y Tecnológica (ANPCYT) and Universidad Nacional de La Plata UNLP, through grants CONICET-IBOL 2318/11, BID-PICT 2012/2524, 2016/2798, 2016/0739 and 11/N852. The funders had no role in study design, data collection and analysis, decision to publish, or preparation of the manuscript.

### Grant Disclosures

The following grant information was disclosed by the authors:
Consejo Nacional de Investigaciones Científicas y Técnicas (CONICET), Agencia Nacional de Promoción Científica y Tecnológica (ANPCYT) and Universidad Nacional de La Plata UNLP: CONICET-IBOL 2318/11, BID-PICT 2012/2524, 2016/2798, 2016/0739 and 11/N852.

### Competing Interests

The authors declare that they have no competing interests.

### Author Contributions

- María G. del Río contributed reagents/materials/analysis tools, prepared figures and/or tables, authored or reviewed drafts of the paper, approved the final draft.
- Analía A. Lanteri analyzed the data, contributed reagents/materials/analysis tools, authored or reviewed drafts of the paper, approved the final draft.

### Data Availability

The specimens examined in the article correspond to the entomological collections cited in the Material and Methods.

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
