# Peer review of "Recognition of species groups of Naupactus Dejean (Coleoptera: Curculionidae) from Argentina and neighboring countries"

_PeerJ, doi:10.7717/peerj.6196_

## Round 0.1 · original submission · Minor Revisions

Please follow the suggestions made by the reviewers, so that your paper can be accepted.

·

Basic reporting

This is an excellent manuscript about weevils of tribe Naupactini, written by the two leading experts in the group. It deals with the highly diverse and economically important South American genus Naupactus, focused on the species that occur in Argentina. This taxonomic contribution allows identification of the nine species groups and provides new, revised and updated information (morphological, biogeographic and on host-plants) for 37 species occurring in Argentina and neighboring countries.
The paper is well written and nicely illustrated. I only have some minor corrections and few suggestions to improve the manuscript:
Line 137. Methathoracic wing, correct to plural “wings”.
Line 171. Please check/correct the sentence where it says “…setae of medium-length hand suberect…”. It is not understandable as it is written.
Lines 240-243. I suggest using present tense here because it is understood that these are nomenclatural changes done in the present paper. Also, placing the word “therefore” (= for that reason, consequently) is not enough because “not being a Naupactini” doesn´t provide justification for being a Tanymecini. I suggest rewording in present tense and providing a justification for your decision, something like:
“…it is not a Naupactini and meets definition of genus Eurymetopellus Emden, therefore it is herein transferred to Tanymecini, genus Eurymetopellus, thus establishing the new combination Eurymetopellus cephalotes (Hustache).”
Line 483. Please check specific epithet of “N. lar (Germar)”.
Lines 845 and 847. There are extra words in the following pairs: “…bears resembles” (delete bears); “…is looks” (I would delete “is”).

Experimental design

no comment

Validity of the findings

no comment

Additional comments

This is an excellent manuscript about weevils of tribe Naupactini, written by the two leading experts in the group. It deals with the highly diverse and economically important South American genus Naupactus, focused on the species that occur in Argentina. This taxonomic contribution allows identification of the nine species groups and provides new, revised and updated information (morphological, biogeographic and on host-plants) for 37 species occurring in Argentina and neighboring countries.
The paper is well written and nicely illustrated. I only have some minor corrections and few suggestions to improve the manuscript:
Line 137. Methathoracic wing, correct to plural “wings”.
Line 171. Please check/correct the sentence where it says “…setae of medium-length hand suberect…”. It is not understandable as it is written.
Lines 240-243. I suggest using present tense here because it is understood that these are nomenclatural changes done in the present paper. Also, placing the word “therefore” (= for that reason, consequently) is not enough because “not being a Naupactini” doesn´t provide justification for being a Tanymecini. I suggest rewording in present tense and providing a justification for your decision, something like:
“…it is not a Naupactini and meets definition of genus Eurymetopellus Emden, therefore it is herein transferred to Tanymecini, genus Eurymetopellus, thus establishing the new combination Eurymetopellus cephalotes (Hustache).”
Line 483. Please check specific epithet of “N. lar (Germar)”.
Lines 845 and 847. There are extra words in the following pairs: “…bears resembles” (delete bears); “…is looks” (I would delete “is”).

Reviewer 2 ·

Basic reporting

The following typographical or grammatical errors should be corrected:

Line 171 – “hand” should be “and”

328 – "withouf" should be "without"

407 – "Cyrtomon-like", with a hyphen, is used elsewhere in the text but not here

There should be consistency with having spaces after numbers and before units. e.g. line 427

738 – "oftern" should be "often"

767 – Gossypium is only spelled with one 's'

835 – "differs" should be used instead of differentiates

Other items that should be addressed:

Line 95 - the specimens borrowed from Charles O'Brien should be reported as being housed at ASUCOB, for Arizona State University Charles O'Brien collection, where the collection is now, in Tempe, Arizona, USA (http://scan-bugs.org/portal/collections/misc/collprofiles.php?collid=121)

Line 169 – there is no couplet 7 in the key! change all 8’s to 7’s (lines 138, 169, 172) and all 9’s to 8’s (lines 175, 176, 179).

Lines 315/387/432/492/571/628/723 – instead of forehead, should be frons or vertex be used? if there is some reason that forehead is preferred please specify.

Experimental design

no comment

Validity of the findings

no comment

Additional comments

This was a very well done study on a large and difficult genus of weevils and will surely be a crucial reference for anyone studying naupactines. There are only very minor errors that should be addressed.

---

## Round 0.2 · accepted · Accept

Thanks for your careful revision.

#